# The Regulation of Intestinal Mucosal Barrier by Myosin Light Chain Kinase/Rho Kinases

**DOI:** 10.3390/ijms21103550

**Published:** 2020-05-18

**Authors:** Younggeon Jin, Anthony T. Blikslager

**Affiliations:** 1Department of Animal and Avian Sciences, College of Agriculture and Natural Resources, University of Maryland, College Park, MD 20742, USA; ygjin@umd.edu; 2Department of Clinical Sciences, Comparative Medicine Institute, College of Veterinary Medicine, North Carolina State University, Raleigh, NC 27607, USA

**Keywords:** myosin light chain kinase, rho-associated coiled-coil containing protein kinase, apical junction complex, tight junction, adherens junction

## Abstract

The intestinal epithelial apical junctional complex, which includes tight and adherens junctions, contributes to the intestinal barrier function via their role in regulating paracellular permeability. Myosin light chain II (MLC-2), has been shown to be a critical regulatory protein in altering paracellular permeability during gastrointestinal disorders. Previous studies have demonstrated that phosphorylation of MLC-2 is a biochemical marker for perijunctional actomyosin ring contraction, which increases paracellular permeability by regulating the apical junctional complex. The phosphorylation of MLC-2 is dominantly regulated by myosin light chain kinase- (MLCK-) and Rho-associated coiled-coil containing protein kinase- (ROCK-) mediated pathways. In this review, we aim to summarize the current state of knowledge regarding the role of MLCK- and ROCK-mediated pathways in the regulation of the intestinal barrier during normal homeostasis and digestive diseases. Additionally, we will also suggest potential therapeutic targeting of MLCK- and ROCK-associated pathways in gastrointestinal disorders that compromise the intestinal barrier.

## 1. Introduction

The single layer of intestinal columnar epithelium forms the body’s largest interface between the external environments in the form of luminal contents [1]. Intestinal epithelial cells allow for the absorption of nutrients and water while providing a physical barrier to harmful components within the gut lumen, including microorganisms and their toxins, from gaining access into subepithelial tissues and the circulatory system [2]. Intestinal epithelial cells are linked by the apical junctional complex (AJC), including tight junctions (TJs) and adherens junctions (AJs) [3]. The AJC, in large part, regulates epithelial barrier function [4]. Intestinal barrier function, and the associated regulatory events centered at AJC regulation, are altered in a number of gastrointestinal disorders, including inflammatory bowel diseases (IBD) [5] and celiac disease [6]. However, there is a lack of well-defined therapeutic or diagnostic targets for AJC in patients with gastrointestinal disorders.

To date, various studies have demonstrated that phosphorylation of myosin light chain II (pMLC-2) is a key regulatory event in the physiological and pathological modulation of AJC [7,8,9,10,11,12,13,14]. Briefly, the perijunctional actomyosin rings are critical to maintaining AJC stability, and contraction of the perijunctional actomyosin ring induces the internalization of AJC proteins. This contraction is largely regulated by pMLC-2 (Figure 1). Myosin light chain kinase (MLCK) and Rho-associated coiled-coil containing protein kinase (ROCK) have been shown to utilize phosphorylation of MLC-2 as a mechanism to regulate the AJC in various forms of intestinal mucosal injury (Figure 2) [7,8,9,10,15,16,17]. However, the mechanistic role of MLCK and ROCK in the regulation of AJC, including TJs and AJs, has not been studied in depth. Therefore, this review focuses on MLCK- and ROCK-mediated intracellular pathways that may play important roles in the regulation of AJC, which in turn is responsible for regulating intestinal barrier function in specific gastrointestinal disorders. Additionally, we review the role of pharmaceutical inhibitors for MLCK or ROCK, suggesting potential therapeutic targeting of MLCK- or ROCK-associated regulation of AJC in gastrointestinal disorders that compromise intestinal epithelial integrity.

## 2. Intestinal Mucosal Barrier Function

The intestinal mucosal barrier supports nutrient and water absorption while preventing the permeation of harmful intestinal luminal contents into the body [18]. Mucosal barriers, including physical barriers and chemical barriers, spatially segregate gut microbiota and the host immune system to avoid unnecessary immune responses to gut microbes [19]. The commensal bacteria, and mucins secreted by the intestinal epithelium, collectively form an important component of the mucosal barrier [20]. As far as the epithelial barrier itself, there are two major selective permeability routes for ions and macromolecules in single-layered epithelium: the transcellular (transepithelial) and paracellular (interepithelial) pathways [21]. The transcellular pathway is associated with the active movement of solutes through transport proteins in the epithelial membrane [22]. The paracellular pathway is associated with the passive movement of water and solutes through the interepithelial spaces [22]. The majority of transmucosal movement of solutes is via the paracellular pathway, and this pathway is regulated primarily by AJC [22]. The paracellular route is composed of two functionally distinct pathways regulated by AJC: pore and leak pathways [18]. The pore pathway is high capacity and charge-selective and allows movement of small ions and uncharged molecules [23,24]. The leak pathway is low-capacity and allows the flux of larger ions and molecules regardless of charge [23,24]. These paracellular pathways are dynamically and precisely regulated by AJC, including TJs and AJs.

## 3. Apical Junctional Complex

Intestinal epithelial barrier function is principally regulated by AJC, which is formed by the TJs and AJs, also contribute to apical-basal cell polarity maintenance and cell signaling events.

### 3.1. Tight Junctions

The TJs are the most apical component of the intercellular junctional complex. Their anatomic structure was initially visualized using transmission electron microscopy by Farquhar and Palade [25]. TJs were identified as regions where the outer leaflets of plasma membranes from adjacent cells appeared to fuse together and seal the intercellular space [25]. However, freeze-fracture microscopy revealed that the TJs are a belt-like structure where the membranes from adjacent cells are in close proximity, and generate zipper-like structures, the so-called TJ strands [26]. These TJ strands consist of multiple protein complexes of transmembrane, cytoplasmic plaque, cytoskeletal, and signaling proteins [27]. At least four different types of transmembrane proteins have been identified at TJs: occludin [28], claudins [29], tricellulin [30], and junctional adhesion molecules (JAM) [31]. Also present within the TJs are the scaffold PDZ domain-expressing zonula occludin (ZO) proteins and cytoplasmic plaque proteins, which serve to form a complex linking transmembrane and cytoskeletal proteins (Figure 1) [23,27]. A number of kinases and signaling proteins, including transcriptional factors (e.g., ZONAB and C/EBP), kinases (e.g., PKC, ERK1, and PI3K), and phosphatases (PP2A and PP1), have also been localized to the tight junction [27,32].

Occludin was the first transmembrane TJ protein to be discovered and is highly expressed in TJ strands [28]. Several studies have reported alteration in barrier function associated with changes in occludin localization, expression, phosphorylation, and ubiquitination. Alternatively, the overexpression of occludin in cultured MDCK cells or fibroblasts increased the number of TJ strands and elevated the transepithelial electrical resistance (TEER), as a measure of barrier permeability to ions [33,34,35]. Caco-2BBe epithelial monolayers with stably knocked down occludin had markedly enhanced TJ permeability as a result of an increased leak pathway [36]. However, occludin appears to be non-essential for TJ barrier formation because occludin-deficient embryonic stem cells are capable of forming TJ strands [37], and occludin knockout mice showed a lack of obvious defects in barrier function [38]. However, more recently, it has been determined that mice deficient in occludin suffer from deafness [39]. Thus, this remains an area in need of further study, as a plethora of in vitro studies suggest that occludin has important functional roles.

Two 22-kDa proteins, claudin-1 and -2, were initially identified in occludin-containing chicken liver junctional fractions by the Tsukita group [29,40]. In mammalian tissues, 27 members of the family have currently been described [41]. The claudin family consists of four transmembrane domains and two extracellular loops that construct TJ strands, which generate a barrier that contains ion and charge-selective pores within the tight junction [42]. Claudins have two different functional subcategories with regard to paracellular permeability called ‘sealing claudins’ and ‘pore forming claudins’ [43]. The ‘sealing claudins’include claudins-1, -3, -5, -9, and -11, and decrease paracellular permeability. For instance, claudin-1-deficient mice showed abnormal TJ barrier formation at the stratum granulosum of the epidermis, leading to loss of water and macromolecules [43]. As a consequence, these mice die of dehydration in the neonatal period [44]. The ‘pore forming claudins’ include claudins-2, -7, -10, -15, -16, and enhance paracellular permeability in a charge and size-selective manner [23]. For example, mice with claudin-2 or claudin-15 deficiencies in the small intestine have revealed that these claudins play a critical role in paracellular channel-like perm-selectivity for extracellular monovalent cations, particularly Na^+^ (i.e., increased movement of Na^+^ via TJ pores) [45,46].

Zonula occludens (ZO, TJP1)-1, -2, and -3 proteins are members of the membrane-associated guanylate kinase family of proteins displaying a characteristic multidomain structure and have shared structural features, including an Src homology 3 domain, a guanylate kinase domain, and an N-terminal region with multiple PDZ (PSD95-Dlg-ZO1) domains [47,48]. These proteins interact together and bind to the cytoplasmic tail of claudins, occludin, and JSMs with their N-termini. ZO proteins interact with the perijunctional actomyosin ring via their C-terminus [49]. ZO-1, -2, and -3 are scaffold proteins that establish numerous protein-protein interactions that cluster at the diverse TJ kinases, phosphatases, small GTPases, and transcriptional factors [32]. More recently, >400 proteins, which directly interact with ZO-1, were identified by mass spectrometry [50]. However, the role of ZO proteins in TJ formation and function is not fully understood. Although ZO-1-deficient cells can maintain the structure of TJs and exhibit normal epithelial barrier properties, the activity of other TJ proteins in assembling TJs is delayed [51,52]. On the other hand, deficiency of ZO-2 or ZO-3 does not affect the formation of TJ in epithelial cells, which suggests that ZO-1 plays a more crucial role in the control of TJ assembly compared to ZO-2 or ZO-3 [51]. In addition to TJ proteins, ZO-1 also binds to the adherens junction proteins afadin and α-catenin [53,54].

### 3.2. Adherens Junctions

The epithelial AJs are positioned below the TJs and are comprised of two families of transmembrane spanning adhesive units: The E-cadherin/catenin family and the nectin/afadin complexes [3,55]. The extracellular regions of E-cadherin and nectin mediate adhesion of cells to their neighbors, while the intracellular regions interact with an array of proteins [3,55]. These intracellular proteins (catenins and afadin) control the assembly and dynamics of AJs by modulating connections with the actin cytoskeleton and stimulating signaling pathways [3,56]. The AJs are dynamic structures that maintain tissue integrity, as well as regulate cell shape and translate actomyosin-generated forces throughout the epithelial tissue [57,58,59]. They also play a major role in embryonic morphogenesis, in addition to the formation and homeostasis of epithelial tissues [57,60].

E-cadherin is a transmembrane glycoprotein that mediates Ca^2+^-dependent intercellular adhesion with a conserved cytoplasmic tail and with an extracellular domain formed of 5 cadherin repeats that mediate homophilic binding with cadherins from adjacent cells [61,62]. The juxta-membrane portion of the E-cadherin cytoplasmic domain associates with p120-catenin, whereas its C-terminal part binds to β-catenin or plakoglobin. β-catenin has been suggested to interact with α-catenin, which harbors an F-actin-binding motif (Figure 1) [63,64].

Nectin is a Ca^2+^-independent immunoglobulin-like cell-cell adhesion molecule. Its extracellular domain contains three Ig-like loops, followed by a single-pass transmembrane domain, and a cytoplasmic tail [65,66]. Like cadherin, nectins mediate cell-cell adhesion and facilitate the establishment of apical-basolateral polarity by nectin dimerization between neighbor cells [65,66]. On the inside of the cell, the nectins bind to numerous cytoplasmic proteins, including afadin. Afadin is an intracellular actin-binding protein and anchors the nectins to the actin cytoskeleton. It also binds α-catenin, ponsin and zonula occludens-1 (ZO-1) (Figure 1) [67]. Nectin also associates with E-cadherin through their respective peripheral membrane proteins, afadin, and catenins, which connect nectin and cadherin to the actin cytoskeleton, respectively. The nectin-afadin interaction is essential for adherens junction maturation, as loss of afadin delays cadherin localization to cell-cell junctions and weakens AJs [68,69].

## 4. Defects of the Apical Junctional Complex in Gastrointestinal Disorders

Alteration of intestinal barrier function is a common component of a number of gastrointestinal disorders, including inflammatory bowel diseases (IBD) and celiac disease [3,70,71]. Disruption of barrier function causes loss of fluid and electrolytes from the body with associated diarrhea, bacterial translocation, and transepithelial migration of neutrophils across the injured epithelium [18]. Although it has not yet been determined whether the loss of barrier integrity is the cause or consequence of these diseases, the intestinal barrier is critical to the pathogenesis of these diseases. This is because abnormal delivery of luminal antigens to the mucosal immune system and infiltration of inflammatory cells may perpetuate the host defense response, leading to the chronicity of the intestinal barrier dysfunction, and increasing disease severity [72,73,74]. Therefore, it is imperative to understand the factors that contribute to the loss of barrier integrity under pathologic conditions.

### 4.1. Inflammatory Bowel Diseases

IBD, including ulcerative colitis (UC) and Crohn’s disease (CD), is associated with intestinal inflammation and chronic-relapsing diarrhea [75,76]. In 2015, approximately 3 million US adults (1.3% of the population) reported being diagnosed with IBD, which was a marked increase of 1 million patients as compared to 1999 (0.9% of the population) [77,78]. Although significant advances have been made in understanding the pathophysiology of IBD, we are far from fully understanding the etiologic pathways that lead to this chronic debilitating disorder. Intestinal barrier dysfunction is believed to be a critical factor in the pathogenesis of IBD because increased paracellular permeability has been identified in both active and quiescent disease states of IBD, and increased permeability results in disease relapse in patients with CD [79]. Both histochemical techniques and expression studies have shown that the composition of the AJC and the localization of individual components are affected in IBD, and account for morphological and functional changes of the epithelial barrier integrity. For instance, claudin-2, a pore-forming tight junction protein, was significantly upregulated in CD [80] and UC [81] patients. A marked downregulation of several junctional proteins, including occludin, ZO-1, E-cadherin, has been noted in CD and UC patients as compared to that of normal subjects or non-inflamed CD and UC epithelium [5,82,83,84]. In addition, other catenins within the AJs, such as α-catenin, β-catenin, and p120-catenin had significantly reduced expression in ulcerated IBD mucosa compared to normal areas [82,84,85]. β-catenin is a critical component of the Wnt signaling pathway, and this pathway acts as the central regulator of epithelial cell homeostasis [86,87]. Wnt ligands induce the stabilization of the transcription co-factor β-catenin in the nucleus, together with Transcriptional factor/Lyhmphoid enhancer-binding factor (TCF/LEF) type transcription factors, and enhance the expression of target genes to maintain intestinal crypt homeostasis [88]. Although the role of membrane-bound E-cadherin: β-catenin complex is under debate, recent studies indicated that Wnt pathway activation could disrupt this complex to increase cytoplasmic β-catenin and further activate Wnt signaling [89]. In addition, E-cadherin can sequester β-catenin in the cytoplasm [90]. In endothelial cells, the role of vascular endothelial (VE)-cadherin and β-catenin in expression of claudin-5 was studied [91]. In the absence of VE-cadherin, the β-catenin-Tcf-4 complex binds to the promoter of the *claudin-5* gene and reduces its expression [91]. This study also supports the importance of the stability of the AJ structure to prevent the activation of β-catenin signaling. Reorganization of AJC proteins has been shown to be mediated by cytokines in inflammatory disease, including tumor necrosis factor-α (TNFα), interferon-γ (IFNγ), lymphotoxin-like inducible protein that competes with glycoprotein D for herpes virus entry on T cells (LIGHT), and IL-1β [12,92,93]. AJCs are bound to the perijunctional actomyosin ring, and disruption of AJC by these cytokines is also closely associated with rearrangement of perijunctional rings induced by increased pMLC-2 [12,92,93]. The expression of pMLC-2 is dramatically elevated in patients with UC and CD, and correlates with colitis activity [94]. Thus, the role of the pMLC-2 regulators, MLCK and ROCK, needs more in depth studies to improve our understanding of the pathogenesis of IBD (Figure 2).

### 4.2. Celiac Disease

Celiac disease is one of the most prevalent chronic autoimmune diseases caused by the activation of innate and adaptive immune responses as a consequence of the ingestion of dietary gluten (specifically gliadin peptides) in genetically susceptible people [95]. Celiac disease is characterized by the presence of the HLA class II molecules (HLA-DQ2 or HLA-DQ8) in antigen-presenting cells and the presence of circulating immunoglobulin A auto-antibodies to transglutaminase [96]. Deamidation by transglutaminase of gluten peptides on gut mucosa increases their immunoreactivity and binding affinity to HLA-DQ2 or HLA-DQ8 [97]. This strongly induces the activation of CD4^+^ T cells that secrete Th1 cytokines such as IFN-γ, IL-4, IL-5, and TNF [98]. The gluten peptides also induce hyperactivation of matrix metalloproteinases and exaggerated enterocyte apoptosis. Recent studies have suggested that the intestinal mucosal barrier dysfunction is a critical factor of the pathogenesis in celiac disease [99,100,101]. In celiac disease, permeability is elevated in the gastro-duodenum and small intestine as a result of disrupted AJC ultrastructure [102]. In active celiac disease, the reduction of ZO-1 phosphorylation makes it unable to link with occludin and to localize at the apical end of the lateral membrane associated with F-actin disorganization. This becomes evident on confocal microscopy, in which the TJ complex was shown to be disrupted [103]. Similarly, the extensive phosphorylation of β-catenin found in mucosa from patients with celiac disease leads to a reduction of the level of co-immunoprecipitated E-cadherin: β-catenin. This is accompanied by disassembly of the AJ complex, and a consequent increase of the β-catenin cytoplasmic pool, which can be confirmed by confocal microscopy [104]. In this scenario, it is likely that dramatic reductions of ZO-1 phosphorylation and excessive phosphorylation of β-catenin impair the interaction of these proteins with their natural partners, occludin and E-cadherin, respectively. This then leads to disruption of AJC and increased intestinal permeability. In another study, mRNA and protein expression of β-catenin was significantly elevated, and nuclear β-catenin extended further up in the crypt (relative to the lumen) in untreated celiac disease patients compared to samples from treated celiac patients and heathy controls [105]. Since the Wnt/β-catenin pathway is critical to intestinal homeostasis in the pathogenesis of celiac disease [106,107], the role of β-catenin in the AJC also needs to be studied in depth in the pathogenesis of celiac disease.

## 5. Role of MLCK in Phosphorylation of MLC-2 to Regulate Intestinal AJC

MLCK is a Ca^2+^-calmodulin-activated serine/threonine kinase that dynamically regulates actomyosin reorganization and cell contraction in smooth, cardiac, and skeletal muscle as well as in non-muscle cells [108,109]. MYLK2 encodes the skeletal MLCK and MYLK3 encodes a cardiac-specific MLCK [110,111]. The MYLK1 gene encodes long non-muscle isoforms (210KDa), short smooth muscle isoforms (108KDa), and telokin (21KDa) that lacks enzymatic activity [110,112,113]. In the intestinal epithelium, two long non-muscle isoforms, MLCK1 (full-length long MLCK) and MLCK2 (which lacks a single exon), are predominantly expressed, and play a critical role in modulating various cell functions [114]. These variants are generated by alternative splicing, have distinct subcellular localizations and functions, and their expression is differentially regulated during epithelial differentiation [114]. MLCK1 is dominantly expressed in villous epithelium, where it is concentrated within the perijunctional actomyosin ring, whereas MLCK2 is expressed throughout the crypt-villus axis [115]. MLCK1 regulates barrier function via phosphorylation of MLC, and selective MLCK1 knockdown decreases AJC permeability [114]. Previous work had determined that MLCK1 expression correlates with a critical physiological regulation in intestinal epithelial cells, and is responsible for Na^+^-nutrient cotransport-dependent TJ regulation [114]. MLCK phosphorylates MLC at threonine-18 and/or serine-19 leading to perijunctional actomyosin ring contraction with disrupted TJ and AJ proteins within the AJC (Figure 1) [116]. The nuclear trans-localized β-catenin induced by IL-1β can also act as a transcriptional repressor for the Claudin-5 gene in endothelial cells [117]. In this study, the non-muscle MLCK regulates β-catenin activation, and nuclear translocation to regulate the IL-1β-induced claudin-5 repression [117]. Taken together, MLCK-dependent MLC phosphorylation is an essential intermediary pathway in physiological AJC regulation.

## 6. Role of MLCK in Gastrointestinal Disorders

Epithelial dysfunction characterized by increased permeability in gastrointestinal disease initiation and progression is promoted by MLCK-dependent intestinal epithelial AJC modulation [22,115,118]. IBD, including UC and CD, is characterized by chronic gastrointestinal inflammation and is associated with significantly impaired barrier function [83]. AJC dysfunction leads to disruption of intestinal integrity, which allows the passage of harmful luminal contents. Tumor necrosis factor (TNF)-α, a proinflammatory cytokine central to IBD pathogenesis, causes intestinal AJC barrier dysfunction-induced increased intestinal permeability via MLCK activation [119]. TNF-α increases MLCK1 (villus) trafficking to the perijunctional actomyosin ring, but has little effect on MLCK2 (crypt-villus) distribution. Despite the unique role of MLCK1 in tight junction regulation, MLCK1 and MLCK2 transcription appear to be activated by TNF-α [120]. In mice with a knockout of intestinal epithelial long MLCK, reduced phosphorylation of MLC-2 resulted in decreased intestinal permeability [114]. Mice treated with MLCK inhibitors also enhanced intestinal barrier function, associated with reduced phosphorylation of MLC-2 [121]. In tissues examined from intestinal resections and biopsies, ileal epithelial MLCK expression was slightly increased in patients with inactive CD. Expression was further elevated during active CD, correlating positively with histological evidence of disease activity [94]. In conclusion, these studies provide key insights into the regulation of MLCK by inflammatory cytokines and the role of MLCK in IBD-associated epithelial barrier loss. Several studies have also shown that elevated MLCK functions have a crucial role in pathogenesis of various digestive diseases associated with ‘leaky’ gut such as celiac disease, irritable bowel syndrome, and ischemia/reperfusion injury [17,122].

## 7. Potential of MLCK Inhibition as a Therapeutic Approach

The MLCK pharmacological inhibitors include ML-7 and ML-9. ML-9 is a classic MLCK inhibitor with an IC_50_ of 3.8 μM, and found to inhibit both Ca^2+^-calmodulin-dependent and –independent smooth muscle MLCK [123]. Another MLCK inhibitor, ML-7, is a membrane-permeable agent which is 30-fold more potent than that of ML-9 (IC_50_ = 300 nM) [124]. Both ML-7 and ML-9 can restore disrupted AJ proteins E-cadherin [125,126] or β-catenin [127]. However, these MLCK inhibitors are not useful for therapeutic purposes because they inhibit many kinases at concentrations necessary to block MLCK. Thus, an MLCK specific inhibitor was developed as a cell-permeant peptide Arg-Lys-Lys-Tyr-Lys-Tyr-Arg-Arg-Lys (designated peptide 18 or Peptide Inhibitor of Kinase; PIK) with amino acid substitutions that provided for an increased peptide selectivity to MLCK [128].

Although inhibitors with specificity are available, these are also unsuitable, as they cannot distinguish between epithelial long MLCK (MLCK1) and smooth muscle MLCK, whose catalytic domains are derived from a single gene and are, therefore, both targeted by inhibitors [110]. Thus, although epithelial MLCK is an attractive therapeutic target, systemic toxicity associated with smooth muscle MLCK inhibition limits the utility of this approach. More recently, however, Turner et al. reported an alternative strategy for the therapeutic inhibition of non-muscle MLCK-dependent barrier loss [129]. They identified a small molecule termed divertin that binds to IgCAM3 and prevents stimulus-induced MLCK1 recruitment to the perijunctional actomyosin ring [129]. The divertin does not interfere with MLCK enzymatic activity, epithelial wound repair, or smooth muscle contraction [129]. However, it prevents MLCK-mediated intestinal barrier loss in vitro and in vivo, restores barrier function in spontaneous colitis, and attenuates experimental, immune-mediated colitis [129].

## 8. Role of Rho/ROCK Signaling Pathway in Intestinal AJC

The Rho family of GTPases is a family of small signaling G proteins and members of the Ras superfamily [130]. The Rho GTPases have been shown to regulate a wide spectrum of intestinal epithelial cell function [130,131]. Of the Rho family, RhoA, Rac, and Cdc42 have been most extensively characterized and play a central role in controlling actomyosin dynamics, as well as other biological processes including focal adhesion, gene transcription, cell cycle, and vesicular transport [132]. Specifically, RhoA promotes the formation of F-actin stress fibers to form and regulate cellular focal adhesion by restructuring the cytoskeleton in response to extracellular stimuli (e.g., growth factor) [133]. Rho GTPases function as a molecular switch and execute their function by switching from the GDP-bound state (inactive) to the GTP-bound state (active). As with many GTPases, the GTP- and GDP-bound states are controlled primarily by three classes of regulatory molecules [134]. Specific guanine nucleotide-exchange factors (GEFs) facilitate the exchange of Rho GTPase associated GDP with GTP, thereby activating the GTPase and resulting in effector binding. On the other hand, GTPase-activating proteins (GAPs) increase the intrinsic rate of GTP hydrolysis, increasing an inactive form with GDP binding [135]. Additionally, guanine nucleotide dissociation inhibitors (GDIs) control the access of Rho GTPases to regulatory GEFs and GAPs, and access to membranes where such effector targets reside [135].

GAPs, GEFs, and GDIs are highly expressed in the intestinal epithelium and are activated by extracellular stimuli, including inflammatory cytokines, growth factors, and bacterial products [134,136]. ROCKs, which are downstream effectors of the GTP-binding Rho proteins, regulate critical intestinal epithelial pathways via regulation of perijunctional actomyosin ring dynamics [8,137]. Two isoforms of ROCKs have been extensively studied: ROCK1 and ROCK2. ROCK1 is widely expressed in non-neuronal tissues, including the liver, lung, and the gastrointestinal tract, whereas ROCKII is principally expressed in the brain and spinal cord [138]. ROCKs belong to the members of the serine/threonine protein kinases family. They are characterized by their effect on the direct phosphorylation of MLC and inactivation of the myosin-binding subunit of myosin light chain phosphatase (MLCP), which is termed the myosin phosphatase target subunit (MYPT1) [139]. This leads to the accumulation of pMLC and subsequent regulation of cytoskeletal contractility and AJC disorganization [140].

## 9. Role of Rho/ROCK in Digestive Diseases

In chronic inflammatory disorders such as IBD, numerous proinflammatory cytokines, including TNF-α, IFN-γ, IL-1 family members such as IL-1β and IL-13, compromise barrier function by promoting endocytosis of epithelial AJC proteins [83]. Such AJC protein internalization is controlled by Rho-/ROCK-/MYPT-/MLC-mediated contraction of the perijunctional actomyosin ring [134,140]. Increased activation of RhoA/ROCK has been detected in inflamed colonic mucosa from patients with CD and rats with TNBS colitis [141]. Furthermore, the ROCK inhibitor Y-27632 significantly reduced colonic inflammation [141,142]. Epithelial barriers inhibit access of microorganisms to tissue compartments, and pathogens have evolved to exploit this important barrier. Some bacteria, such as *Salmonella enterica* and *Helicobacter pylori*, inject their own proteins into epithelial cells to activate RhoA/ROCK signaling, which in turn compromises TJ structure and function. As an additional example, *Escherichia coli* cytotoxic necrotizing factor1 compromises TJs by increasing Rho activation [143]. Furthermore, lipopolysaccharide, a toxin formed from the outer membrane of Gram negative bacteria, activates RhoA by increasing p115RhoGEF protein levels, which also has the effect of compromising the epithelial barrier [144,145].

In colorectal cancer, altered expression of AJC proteins is a hallmark of disease progression [3]. For example, reduced expression of E-cadherin and p120 catenin is associated with invasion of cancer cells via Rho signaling [146,147]. In cancer cells, p120catenin has also been reported to activate Rac1, which, in combination with reduced substrate adhesion due to RhoA inhibition, promotes cancer cell invasion [148]. Additionally, decreased activation of Cdc42 and Rac1 correlates with the increased invasive potential of several epithelial cell lines derived from metastatic colorectal adenocarcinoma. This phenotype reverts when ROCK is inhibited [149,150].

## 10. The potential of ROCK Inhibition as a Therapeutic Approach

Various small-molecule inhibitors have been developed to investigate the physiological roles of ROCK in several cell types and animal models. Y-27632 and fasudil (HA-1077) have been used as ROCK selective inhibitors, and target ATP-dependent kinase domains of ROCKs [116,117]. Fasudil, a novel isoquinoline sulfonamide derivative, and the only clinically available ROCK inhibitor has been used clinically to prevent cerebral vasospasm after subarachnoid hemorrhage [151,152]. Fasudil inhibits inflammatory responses by controlling the polarization of microglia/macrophages, maintains the integrity of the blood-brain barrier, and influences the function of astrocytes [153]. Y-27632 was identified by its ability to inhibit phenylephrine-induced contraction of a rabbit aortic strip. It has also played a major role in better understanding the physiological roles of ROCK, including cell adhesion, cell motility, vascular and smooth muscle contraction, and cytokinesis [154]. However, these ROCK inhibitors cannot discriminate between ROCK1 and ROCK2 or the role of ROCKs in individual component cells. Furthermore, at higher concentrations, ROCK inhibitors can also inhibit other serine-threonine kinases such as PKA and PKC [155]. However, more recently, the ROCK-1 selective inhibitor dihydropyrimidines and ROCK-2 selective inhibitor SLx-2119 [156] offer the potential for more targeted ROCK inhibition.

## 11. Conclusions

The contraction of the perijunctional actomyosin ring-induced by phosphorylation of MLC-2 is a critical pathway in the regulation of physiologic and pathophysiological internalization of components of the AJC. Therefore, the continued study of the MLC-2 pathway will lead to a greater understanding of the pathogenesis of digestive diseases characterized by increased mucosal permeability, or ‘leaky gut’ [157]. The phosphorylation of MLC-2 is primarily regulated by MLCK and MLCP activity. However, ROCK, which is the main downstream effector of activated RhoA, can also directly phosphorylate MLC-2, as well as phosphorylating the myosin-binding subunit of MLCP, MYPT1, thereby inducing inhibition of MLCP. Thus, the activation of ROCK also results in increased AJC permeability by perijunctional actomyosin ring contraction induced by elevated pMLC-2 (Figure 2) [158]. However, therapeutic approaches for inhibition of intestinal epithelial MLCK or ROCK are not clinically available to date, because MLCK and ROCK pathways are involved in multiple critical cellular signaling events involved in homeostasis. Nonetheless, this may change as more selective inhibitors become available. For instance, an exquisitely specific MLCK inhibitory peptide, PIK, has been developed and tested in TNF-treated intestinal epithelial monolayers [159]. Additionally, the recent identification of divertin has provided the ability to inhibit stimulus-induced MLCK1 recruitment to the perijunctional actomyosin ring, thereby preventing MLCK-induced barrier loss without interference with MLCK enzymatic activity, epithelial cell wound repair, or smooth muscle contraction [129]. However, further basic research is needed to improve our understanding of these complex signaling pathways, and the interactions between them. Ultimately, understating MLCK/ROCK-associated AJC pathways will offer the opportunity to develop new pharmacological compounds with more selective actions on impaired intestinal barriers in patients with gastrointestinal disorders.

## Figures and Tables

**Figure 1 ijms-21-03550-f001:**
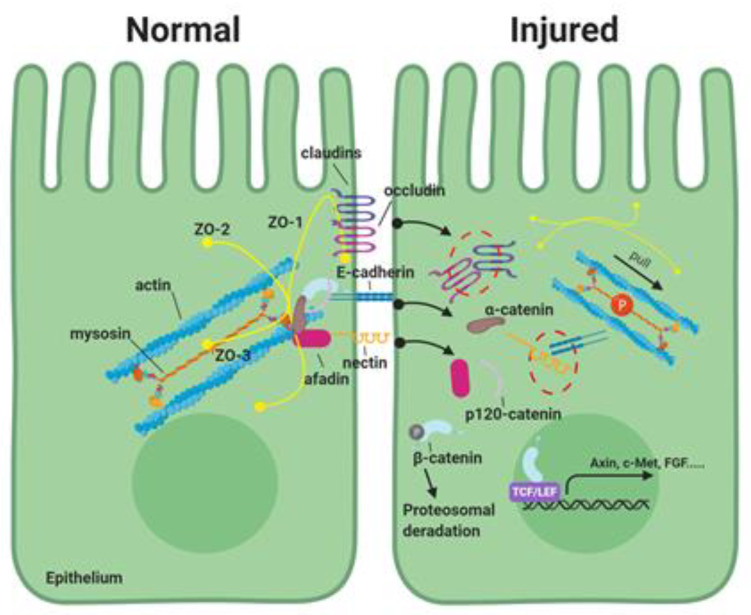
The role of phosphorylation of myosin light chain-2 (MLC-2) in the apical junctional complex in the intestinal epithelial paracellular pathway. The apical junctional complex consists of tight junctions (TJs) and adherens junctions (AJs). TJs consist of transmembrane proteins (e.g., claudins and occludin), and scaffold PDZ domain-expressing zonula occludin (ZO) proteins, which serve to form complex linking transmembrane proteins to the cytoskeletal actomyosin ring. AJs are located below the TJ and are comprised of two families of adhesive units: The E-cadherin/catenin family and nectin/afadin complexes. These proteins are dynamically regulated to maintain epithelial integrity. In intestinal disorders, increased phosphorylation of MLC-2 induces contraction of the perijunctional actomyosin ring and disrupt the apical junctional complex. This results in elevated paracellular permeability during the pathogenesis of various digestive diseases.

**Figure 2 ijms-21-03550-f002:**
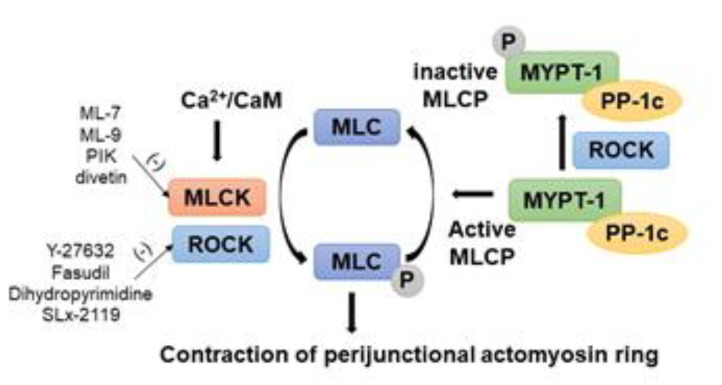
Regulation of phosphorylation of myosin light chain-2 (MLC-2) by myosin light chain kinase (MLCK)/Rho-associated coiled coil containing protein kinase (ROCK) to regulate AJC barrier in intestinal epithelial cells. Paracellular permeability is principally determined by the phosphorylation level of the regulatory light chain of myosin, MLC-2, which is regulated by the enzymes MLCK and ROCK. Increased intracellular Ca^2+^ levels stimulate MLCK activity, which directly phosphorylates MLC-2. Enhanced ROCK activity also directly phosphorylates MLC-2 and inhibits MCL phosphatase (MLCP) activity by phosphorylating the myosin phosphatase target subunit 1 (MYPT1). There are pharmaceutical inhibitors for MLCK and ROCK listed.

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
