# Peer review of "The Regulation of Intestinal Mucosal Barrier by Myosin Light Chain Kinase/Rho Kinases"

_ijms, 2020, doi:10.3390/ijms21103550_

Round 1

Reviewer 1 Report

The revised manuscript satisfies most of the concerns that I originally had with the first submission.  However, Figure 1 was not changed.  The membrane proteins that were internalized by injury are still floating within the cytoplasm, rather than being associated with endosomes.  Was this an oversight?  The authors stated in their response to the reviews that this was changed.  Indeed, the other reviewer recommended that the actin cytoskeleton be shown in association with the apical junctional complex.  Please advise.

Author Response

1. The revised manuscript satisfies most of the concerns that I originally had with the first submission.  However, Figure 1 was not changed.  The membrane proteins that were internalized by injury are still floating within the cytoplasm, rather than being associated with endosomes.  Was this an oversight?  The authors stated in their response to the reviews that this was changed.  Indeed, the other reviewer recommended that the actin cytoskeleton be shown in association with the apical junctional complex.  Please advise.

Response:

1. We thank the reviewer for pointing out this error, and we apologize for uploading the incorrect image. We had revised the image based on reviewer feedback, and have uploaded the correct image with the present revision

Reviewer 2 Report

The authors have improved the manuscript and is worth of pubblication.

Author Response

1. The authors have improved the manuscript and is worth of publication.

Response:

1. We thank the reviewer for critiquing our paper, and finding it to be worthy of publication

This manuscript is a resubmission of an earlier submission. The following is a list of the peer review reports and author responses from that submission.

Round 1

Reviewer 1 Report

The review of Jin and Blikslager „The regulation of intestinal mucosal barrier by myosin light chain kinase/rho kinases“ summarizes the role of MLC-2 for paracellular barrier permeability of the intestine. Phosphorylated MLC-2 leads to contraction of the apical action-myosin ring complex and thus, mechanically increasing the paracellular permeability of the apical junctional complexes. The manuscript gives a good overview about MLC-2, its regulation and role in apical junction integrity.

General remarks and questions.

A detailed description of the actin/myosin ring complex and the signalling pathways activating MLC-2 need to be part of the introduction, since it’s important for the rest of the manuscript to understand how they work.

In the whole manuscript are mainly reviews cited. More research article should be included.

The figure 1 should be improved. The actin/myosin ring complex is just laying somewhere in the cell? The right cell should be drawn as a result of contraction. Just naming “Zos” and “catenins” is a bit simple à ZO-1, 2, 3 and a, b, g-catenin should be included with their subcellular locations. After internalization of E-cadherin or claudins, the outer interaction domain is localized inside the endosomes and the intracellular domains interacting with ZO1 or catenins pointing towards the cytosol. B-Cat and ZO-1 are dissociating intracellularly and at least b-cat gets phosphorylated and degraded or shuttles into the nucleus (under pathologic conditions).

Since beta-catenin is several times mentioned, what is known about beta-catenin and pMLC interactions. (like in Beard RS et al., 2014; DOI: 10.1242/jcs.144550)

The figure 2 could be improved as well. At least the known inhibitor could be included.  

Minor points

line 81. “…most prominently PKC…”. From the literature PKC seems to be not more important as others, PKA for example.

Line 96, 97: Currently known claudins expressed in humans: the reference 37 seems to be not correct. I could not get the paper, but the abstract says nothing about claudins. However, its anyway a bit old for “currently” (2011). There are newer reviews.

Line 110: I like this review, but its from 2004. There is some newer information on PDZ-domains and ZO-proteins. Citation of research papers would be nice as well. By the way, ZO-proteins have a new name, TJP1. Some more information on ZO-1, 2 and 3 interaction with each other would be helpful.

Line 167. Inflammatory bowel diseases. Increased tissue permeability and decreased or increased junctional protein expression is reported in the manuscript. How could MLC2 lead to changed expression? Please describe a bit more the mechanism. Internalization of the junctional complex, degradation of proteins pointing towards the activation of the perijunctional ring complex? The function of beta-catenin needs to be described as well. Maybe using the review of Gavard J and Gutkind JS. VE-cadherin and claudin-5: it takes two to tango, for example.

Line 189. Celiac disease. Again, beta-catenin effect is not explained. The two sentences at the end do not help to include MLC-2 into celiac disease context. If there is no reference for MLC and celiac disease, please remove it.  

Line 259 MCLK: a typing error

Line 276. Role of Rho/ ROCK signalling pathway in intestinal AJC. The title includes Rho/Rock in intestinal AJC. The whole paragraph says nothing about intestinal Rho/Rock effects.

Reviewer 2 Report

The manuscript by Jin and Blikslager reviews reviews the role of myosin light chain kinase and Rho kinase activity in the regulation of the paracellular barrier, particularly in the intestinal epithelium and its dysfunction during disease processes.  The review is useful and interesting.  Overall, this is a strong contribution.  However, there are numerous places that the manuscript can be improved.  I outline my concerns below.

1. There are several places where the manuscript is redundant.  These should be reduced.

2. There are many places where commas are needed.  I realize that many people in science avoid commas in their writing, but they are necessary.  Importantly, there are many run on sentences as a consequence of not using commas (two sentences separated by a conjunction like “and” should have a comma: “, and”).

3. There are numerous factual statements that do not have an accompanying citation.  These should be corrected.

4. Figure 1 shows membrane proteins that appear to be free in the cytoplasm in the injured cell.  The internalization of membrane proteins will occur through endocytosis (membrane traffic).  The figure gives an incorrect view of the injury process.

5. Line 162: The authors state that something is “well accepted”.  It would be better to give examples of the evidence and give a citation rather than claim acceptance.

6. Line 188: The phrase “is need to be more studied” should be corrected (perhaps “needs more study”).

7. Line 198: What does the word “It” refer to in the beginning of the sentence?  Right now it is referring to IL-15.  Please clarify this sentence.

8. Line 205: What is co-immunoprecipitated with E-cadherin?  Please clarify.

9. Line 214: The heading is unclear: “MLCK in pMLC-2” should be clarified.

10. Line 222: The sentence is redundant: “splice variants generated by alternative splicing” should read “variants generated by alternative splicing”.

11. Line 245: The phrase “a genetic absence” is vague.  Knockout?  Mutation?

12. Line 332: The abbreviation of BBB is not needed.  It is used once.  It was also not defined.

13. Figure 2 should be introduced earlier because is serves as a schematic that would guide the reader.